# Machine translationese of large language models: Dependency triplets, text classification, and SHAP analysis

**Shukang Zhang** [ORCID]*, **Chaoyong Zhao**

School of Foreign Languages, East China Normal University, Shanghai, China

* mikeashjd@163.com

## Abstract

This study addresses the challenge of distinguishing human translations from those generated by Large Language Models (LLMs) by utilizing dependency triplet features and evaluating 16 machine learning classifiers. Using 10-fold cross-validation, the SVM model achieves the highest mean F1-score of 93%, while all other classifiers consistently differentiate between human and machine translations. SHAP analysis helps identify key dependency features that distinguish human and machine translations, improving our understanding of how LLMs produce translationese. The findings provide practical insights for enhancing translation quality assessment and refining translation models across various languages and text genres, contributing to the advancement of natural language processing techniques. The dataset and implementation code of our study are available at: https://github.com/KiemaG5/LLM-translationese.

## Introduction

Translation is a complex conversion process, and the resulting text inevitably retains traces of this transformation. Scholars have described how translated language differs from original texts, using terms like "interlanguage", "third code", and "translationese" [1–3]. The emergence of translationese can be attributed primarily to two factors: firstly, the spillover of source language structures and patterns into the target text; and secondly, the general effects of the translation process that are independent of specific source language influences [4]. Additionally, the presence of culture-specific elements in translated texts—such as references to source culture concepts or artifacts—can indicate their translated nature, although these aspects are typically not included in discussions of translationese [5,6].

In recent decades, the focus of translation studies has shifted from the notion of equivalence to a target-text-centered orientation, with the concept of "translationese" evolving to carry a neutral connotation. Currently, it often refers to universal linguistic features in translation that set translated texts apart from original (non-translated)

**Data availability statement:** The minimal anonymized dataset and code necessary to replicate the study findings are publicly available in a stable repository. The data and analysis scripts can be accessed at: https://github.com/KiemaG5/LLM-translationese.

**Funding:** This research was supported by the Shanghai Philosophy and Social Science Planning Fund Project (Grant No. 2023BYY004) awarded to Chaoyong Zhao. The funder's website can be accessed at: http://www.sh-popss.gov.cn/. The funders had no role in study design, data collection and analysis, decision to publish, or preparation of the manuscript.

**Competing interests:** The authors have declared that no competing interests exist.

texts. These features, considered to be more general and independent of specific source or target language influences, are typically referred to as "translation universals" [7]. Such universals encompass phenomena like simplification, normalization, explicitation, and, in some instances, source language interference [8–11].

### Translation universals and inconsistent findings

While the concept of translation universals is widely acknowledged, and scholars concur that translated texts exhibit distinctive linguistic features, empirical findings remain inconsistent, often producing contradictory or inconclusive evidence across different language pairs, genres, and methodologies. Specifically, Baroni and Bernardini [12] emphasize the use of comparable corpora as a methodological approach but acknowledge the difficulty in obtaining truly comparable data, which may introduce confounding variables and unreliable outcomes. Similarly, Liang and Sang [13] observe that relying on simple linguistic metrics like type-token ratio (TTR), lexical density, and word frequencies may obscure the complexity of translated language and fail to consistently identify universal traits. Wang et al. [14] further emphasize that these commonly used metrics often produce divergent results across different language pairs.

### Machine translationese and LLMs

Unlike traditional "translationese", which arises from comparisons between original and translated corpora, machine translationese emerges from comparisons between human translations and machine-generated translations. This reflects distinct patterns in the latter and remains relatively under-explored. This phenomenon is often attributed to algorithmic bias, where models tend to replicate high-frequency patterns in the training data, resulting in linguistically less varied and more homogenized outputs. Earlier studies on Statistical Machine Translation (SMT) and Neural Machine Translation (NMT) systems have consistently demonstrated these characteristics of machine translationese [15,16].

In recent years, the emergence of Large Language Models (LLMs) has significantly influenced translation studies and opened up new avenues for exploration. LLMs exhibit remarkable capabilities in language generation tasks that require deep linguistic understanding and creativity, with machine translation being a prime example of their strengths [17]. In high-resource language pairs, LLMs often outperform specialized NMT and SMT systems, which is largely attributed to their exposure to vast amounts of training data and their ability to leverage extensive linguistic patterns effectively [17–19]

Despite the advanced modeling capabilities of LLMs, questions remain about whether they have fully addressed the challenges associated with machine translationese. Specifically, it is still relatively unclear if their outputs continue to exhibit reduced lexical and morphological richness or if we can still distinguish between human and LLM-generated translations. Recent work by Sizov et al. [20] has contributed significantly to this area by extending the investigation of machine translationese to LLMs, distinguishing their outputs from human translations through embedding-based classification methods. Similarly, Vanmassenhove [21] explored societal biases, such as

gender bias, in the context of Machine Translation, though their study focused less on syntactic granularity and more on the broader implications of bias in translation technologies. These studies collectively highlight the importance of further research that combines advanced LLM analysis with linguistically interpretable frameworks. Such approaches, like the dependency-based features employed in the current study, are valuable for comparing LLM-generated and human translations.

## Dependency syntax

Previous research on dependency syntax has delved into two key metrics: dependency distance (DD) and dependency direction (DDir). Building upon foundational works [22–24], recent studies have focused on DD as a critical measure of syntactic complexity and cognitive load. Research in this domain can be broadly categorized into three aspects.

(1) Dependency Distance Minimization (DDM) suggests that languages tend to shorten the distance between grammatically linked words to make sentences easier to process. Research shows this pattern exists across many languages [25], hinting at a common cognitive strategy.

(2) Cross-Linguistic Comparisons: These comparisons highlight typological differences, as outlined by Bi and Tan [26]. They observed that head-final languages, such as Japanese, Korean, and Turkish, have longer mean dependency distances (MDD) due to their structural and typological features.

(3) Applications in Various Linguistic Contexts: Recent applications of DD span interpreting, L2 academic writing, and constrained bilingual production. Xu and Liu [27] examined how interpreting directionality influences syntactic strategies, Bi and Tan [26] investigated native language transfer in L2 writing, and Ma et al. [28] identified similarities between translated English and non-native English. These studies demonstrate DD's utility in revealing cognitive load and syntactic difficulty across diverse linguistic contexts.

On the other hand, Dependency Direction (DDir) examines the relative ordering of governing and dependent words in syntactic constructions. Originating from Tesnière's [29] foundational work and further developed into a quantitative framework [30], DDir has provided valuable insights into language typology. By analyzing head-initial and head-final patterns, DDir research has not only contributed to the classification of world languages along a continuum, facilitating both diachronic and synchronic comparative studies [31,32], but also has been recently applied to diverse linguistic contexts, such as translation and second language writing. For example, Ma et al. [28] explored how translated texts may exhibit differing dependency direction patterns compared to original compositions, while Liang & Sang [13] investigated the influence of native language structures on second language writing.

However, metrics such as MDD and the percentage of head-final or head-initial dependencies, while useful, often average and normalize values across all dependency types. This can mask finer syntactic variations within texts by overlooking distinctions among different parts of speech and dependency relations. Consequently, these measures may not fully capture the complexity and diversity of syntactic structures. To address this limitation, the current study adopts the dependency triplet features proposed by Hu et al. [33]. Each triplet feature integrates three elements: a dependency relation, the part of speech (POS) of the dependent word, and the POS of the governing word. With 10 types of dependency relations and 10 types of POS categories, this approach can theoretically generate up to 1,000 distinct dependency triplet features, leading to each translation sample being represented as a high-dimensional feature vector. Such a detailed feature set allows for a nuanced representation of translations, providing a rich foundation for computational modeling.

## Text classification

As discussed in Section Machine translationese and LLMs, identifying machine translationese in LLM-generated translations is typically framed as a classification problem, aiming to distinguish human translations from machine-generated ones. This classification paradigm is rooted in earlier translationese studies, where machine learning methods such as

Support Vector Machine (SVM) have been widely employed [12,34–36]. While these methods effectively validated hypotheses about translationese, their reliance on handcrafted, fragmented features—like n-grams or type-token ratios—posed limitations. These features often fell short in capturing the holistic nature of translationese traits, particularly with the emergence of LLM-generated translations, which exhibit complex and nuanced stylistic markers. This underscores the necessity for more sophisticated and comprehensive feature representations, such as the dependency triplet features, to thoroughly analyze and differentiate machine translationese from human translation outputs.

Recent studies have explored multi-model frameworks for classifying machine translationese [14,37]. Both studies utilized entropy-based metrics as key features—Liu et al. focused on character, wordform n-gram and POS-level entropy, while Wang et al. incorporated syntactic rule entropy derived from syntactic trees. However, these features represent only certain aspects of syntax, without fully capturing the broader syntactic complexity or linguistic nuances inherent in (machine) translationese. Furthermore, more recent research has also employed deep learning methods, utilizing architectures like BERT and Llama to address the task of distinguishing translationese, as seen in studies by Amponsah-Kaakyire and Sizov [20,38]. These methods leverage embedding-based representations, demonstrating marked improvements in classification performance. However, these improvements often stem from topic-related differences between corpora rather than capturing the essence of (machine) translationese per se.

These gaps underscore the necessity for a comprehensive and linguistically grounded approach. Traditional studies often grapple with fragmented feature sets, while more recent deep learning methodologies lack explicit linguistic interpretability in their decision-making processes. To address these challenges, this study employs 16 machine learning classifiers with comprehensive dependency triplet features that capture holistic syntactic patterns across multiple linguistic levels. These linguistically interpretable features enable systematic analysis of translation characteristics while maintaining methodological rigor through 10-fold cross-validation, and thus providing robust discrimination between human and LLM translations.

## Model interpretability

While achieving high classification accuracy is important, the ability to interpret and explain model decisions is equally crucial. Explainability in translationese classification has been explored through feature-engineering and neural network-ased methods. As summarized by Amponsah-Kaakyire et al. [38], feature-engineering approaches prioritize quantifying the importance of handcrafted feature, using techniques like inspecting model weights, calculating information gain to assess the relevance of each feature to the outcome, and conducting ablation studies to observe changes in performance when specific features are removed. These techniques help provide insights into which linguistic features are significant for distinguishing translationese.

Neural methods, though less extensively explored, have begun incorporating strategies for improving explainability. For instance, Pylypenko et al. [39] utilized handcrafted features to explain neural predictions, offering a bridge between traditional and modern approaches. Moreover, Amponsah-Kaakyire et al. [38] applied Integrated Gradients [40], a technique that attributes importance to inputs by integrating gradients along a path from a baseline to the input. These integral techniques pave the way for more explainable models, balancing the sophistication of neural networks with the interpretability of their decisions.

SHAP provides a more robust framework for interpreting machine learning models [41] and has been underutilized in translationese studies. Liu et al. [37] employed SHAP to analyze global feature importance but did not explore its potential for revealing local contributions or feature interactions. In this study, SHAP's interpretive power is harnessed to analyze dependency triplet features, offering a versatile approach to understand the syntactic traits of LLM-generated translations. By leveraging SHAP, the study not only improves the interpretability of model predictions but also provides deeper insights into the specific syntactic traits that set apart machine translationese.

## Research gaps and research questions

Despite significant progress, key gaps persist in understanding the linguistic characteristics of translated texts, particularly with respect to machine translationese and LLMs. This study addresses these challenges through four focused strategies:

(1) Controlling External Variables: By comparing multiple human and LLM-generated translations of the same novel, the study ensures that all translations originate from a single source text. This approach helps maintain thematic and stylistic consistency, allowing for a clearer analysis of linguistic differences attributable to the translation method only.

(2) Enhanced Syntactic Analysis: Moving beyond basic metrics such as type-token ratio and dependency distance, the study employs an enriched syntactic analysis. This involves combining dependency relations with POS tags to capture more nuanced syntactic patterns that distinguish between different types of translated texts.

(3) Interpretability with SHAP: The study uses SHAP to achieve interpretability at both global and local levels. SHAP provides insights into how specific syntactic features derived from dependency parsing contribute to distinguishing LLM-generated translations from human ones. By illuminating these features, the study enhances our understanding of the distinct linguistic traits of machine translationese.

Based on these strategies, the study is structured around the following key research questions:

(1) **Classification Performance**: To what extent can the 16 models effectively discriminate between human translations and LLM translations based on dependency triplet features?

(2) **Feature Importance**: What are the most significant dependency triplet features in differentiating human from LLM translations, and what patterns emerge from the SHAP analysis of these features?

## Corpus

This study utilizes an expanded parallel corpus derived from the novel *Bian cheng* (*Border Town*), authored by Shen Congwen (沈从文), a master of Chinese contemporary literature and a nominated candidate for the Nobel Prize in Literature. The corpus encompasses the source text and eight translations: four produced by human translators and four generated by LLMs. The machine translations include two iterations from Baidu's ERNIE—specifically ERNIE-3.5 and ERNIE-4—and two iterations from OpenAI's ChatGPT—namely, ChatGPT-3.5-Turbo and ChatGPT-4. Table 1 presents an overview of the corpus employed in this study.

The source text, *Bian cheng*, was sourced from *The Complete Works of Shen Congwen* (《沈从文全集》) as edited by Zhang [42]. With approximately 50,000 Chinese characters, this novella provides adequate depth for thorough analysis while remaining sufficiently concise to facilitate batch translation via the API interfaces of LLMs.

**Table 1. Overview of the *Bian cheng* Parallel Corpus.**

| Version | Author/Translator(s) | Publisher | Year | Word Count |
|---|---|---|---|---|
| Source Text | Shen Congwen | Beiyue Wenyi Publisher [42] | 1934 | 49, 045* |
| *Green Jade and Green Jade* | Hahn, E., & Shing, M. | T'ien Hsia Monthly [43] | 1936 | 41, 611 |
| *The Frontier City* | Ching, T., & Payne, R. | George Allen & Unwin, Ltd. [44] | 1947 | 36, 006 |
| *The Border Town* | Yang, G. | Panda Books [45] | 1981 | 28, 755 |
| *Border Town: A Novel* | Kinkley, J., C. | HarperCollins [46] | 2009 | 38, 481 |
| *ERNIE-3.5-Version* | ERNIE-3.5 | N/A | 2024 | 35, 902 |
| *ERNIE-4-Version* | ERNIE-4 | N/A | 2024 | 35, 824 |
| *ChatGPT-3.5-Turbo-Version* | ChatGPT-3.5-Turbo | N/A | 2024 | 37, 099 |
| *ChatGPT-4-Version* | ChatGPT-4 | N/A | 2024 | 38, 303 |
| **Parallel Units** | | | | **97** |
| **Translations per Unit** | | | | **8** |
| **Total Translation Samples** | | | | **97×8=776** |

**\*Chinese characters.**

Initially, the corpus was constructed by aligning the source text with the four human translations at the sentence level, resulting in a 1-to-4 parallel structure. In this format, each original sentence was paired with its corresponding human translations. To further expand the corpus, these original sentences were subsequently input into the APIs of the four LLMs using a standarized prompting strategy. The employed prompt template was as follows: "This is a Chinese to English literary translation task. Please provide the English translation for these sentences: {ST}." This approach ensured consistency in the translation process across all models, culminating in a 1-to-8 parallel corpus, wherein each original sentence was paired with eight translations—four human-generated and four machine-generated.

However, during the analysis, it became apparent that individual sentences provided limited contextual information, thereby hindering the ability to capture more complex linguistic patterns and stylistic features. To address this issue, the corpus was restructured into larger units. Specifically, 20 consecutive aligned sentences were grouped into a single unit, resulting in the creation of 97 parallel units. Each unit now contains 8 parallel translations for the same set of 20 sentences.

## Feature engineering

This study utilizes dependency triplets as syntactic features, grounded in dependency grammar, which characterizes binary relations between linguistic units and identifies asymmetrical relationships between a governor and its dependent, along with specific grammatical relation labels [24]. Based on these foundations, this study implements a structured feature representation format (dependency, dependent_POS, governor_POS) that encapsulates both structural relationships through dependency labels and lexical-grammatical properties through POS tags, following the methodology established by Hu et al. [33].This approach provides a comprehensive and interpretable framework for analyzing the linguistic characteristics of translation texts.

## Dependency triplet extraction

Fig 1 illustrates examples of dependency triplet extraction. For the two example sentences, "ECNU is excellent" and "ECNU has two campuses", the extracted dependency triplets include the following:

- **nsubj_PROPN_AUX/VERB**: The subject is a proper noun ("ECNU"), while the predicate is either an auxiliary verb ("is") or a main verb ("has").

- **ROOT_AUX/VERB_AUX/VERB**: The root of the sentence is either the auxiliary verb "is" or the main verb "has".

- **acomp_ADJ_AUX**: The adjective "excellent" functions as an adjectival complement to the auxiliary verb "is".

- **nummod_NUM_NOUN**: The numeral "two" functions as a modifier to the noun "campuses".

- **dobj_NOUN_VERB**: The noun "campuses" functions as the direct object of the main verb "has".

Table 2 demonstrates the process of vectorizing the two sentences using dependency triplet features. In the table, each column represents a sentence, and each row corresponds to a specific syntactic feature. The values in the table illustrate the presence or absence of a feature in the sentence, using binary for simplicity. However, in practice, these values denote the absolute frequency of the features, which are subsequently normalized according to the length of the translation sample to ensure comparability across different translation samples. Additionally, a feature reduction step was applied to emphasize the most frequent and representative dependency triplets, as discussed in section Feature extraction and dimensionality reduction.

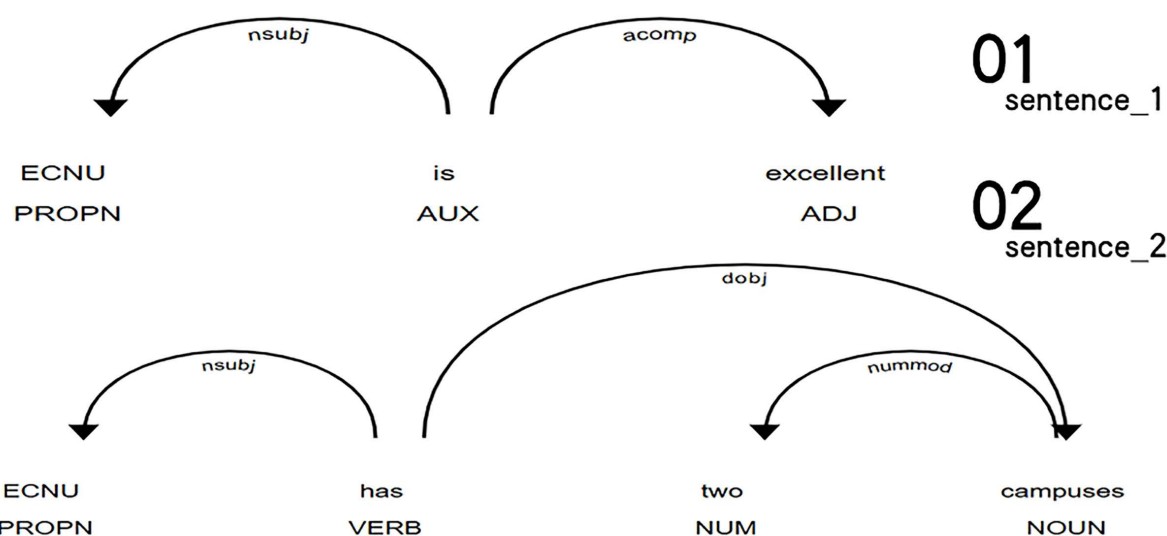

**Fig 1. Visualization of Dependency Triplet Extraction.**

**Table 2. Illustrative Dataset for Sentence Vectorization.**

| Feature | Sentence 1 | Sentence 2 |
|---|---|---|
| nsubj_PROPN_AUX | 1 | 0 |
| ROOT_AUX_AUX | 1 | 0 |
| acomp_ADJ_AUX | 1 | 0 |
| nsubj_PROPN_VERB | 0 | 1 |
| ROOT_VERB_VERB | 0 | 1 |
| nummod_NUM_NOUN | 0 | 1 |
| dobj_NOUN_VERB | 0 | 1 |

## Dependency parsing implementation

The technical implementation utilizes spaCy's English language model en_core_web_sm [47]. The model achieves a labeled attachment score (LAS) of 89.92% and an unlabeled attachment score (UAS) of 91.77%, where LAS measures the percentage of tokens that are assigned both the correct syntactic head and dependency label, while UAS considers only the correctness of the syntactic head assignment. Additionally, the model demonstrates high accuracy in POS tagging (97.29%), indicating reliable identification of grammatical categories, as reported in the official release documentation for en_core_web_sm-3.8.0.

## Feature extraction and dimensionality reduction

In the corpus, each non-punctuation token is associated with a (dependency, dependent_POS, governor_POS) triplet, resulting in 286,849 triplet instances initially. From these, 2,352 unique triplet types were identified. To focus on the most frequent and representative patterns, a frequency threshold of 500 occurrences was applied, reducing the feature space to 108 high-frequency triplet types. Remarkably, these 108 types represent 227,389 triplet instances, accounting for 79.3% of all instances in the corpus. This approach ensures both computational efficiency and substantive coverage of key syntactic phenomena.

The final dataset consists of 776 translation samples, each represented as a 108-dimensional feature vector of normalized frequencies for the selected high-frequency triplet types. Each sample is paired with a binary label (0 for human-translated, 1 for LLM-translated), resulting in a total of 109 columns in the dataset.

## Methodology

### Text classification

To comprehensively evaluate the discriminative capability of dependency triplet features across different algorithmic paradigms, We employed 16 machine learning classifiers spanning eight categories: tree-based models, boosting algorithms, linear models, probabilistic classifiers, discriminant analysis, support vector machines, nearest neighbors, and neural networks. All implementations followed the default settings using scikit-learn [48], with tree-based boosters adopting their native libraries, as we aimed to test whether our dependency triplet feature set could reliably distinguish translation types across different classifier architectures without model-specific optimizations. To ensure robust and reliable evaluation, all classifiers were assessed using 10-fold cross-validation, with both mean F1-score and standard deviation recorded to enable robust performance comparison.

### Model performance estimation

To comprehensively evaluate model performance, several complementary metrics are employed based on the confusion matrix shown in Table 3.

Based on these fundamental counts, we calculate several performance metrics. Accuracy measures the overall proportion of correct predictions:

$$Accuracy = \frac{TP + TN}{TP + TN + FP + FN} \tag{1}$$

Precision measures the proportion of correct positive predictions among all positive predictions:

$$Precision = \frac{TP}{TP + FP} \tag{2}$$

Recall measures the proportion of actual positive cases that were correctly identified:

$$Recall = \frac{TP}{TP + FN} \tag{3}$$

The F1-score provides a balanced measure between precision and recall:

$$F1 = 2 \times \frac{Precision \times Recall}{Precision + Recall} \tag{4}$$

**Table 3. Confusion Matrix for Translation Classification Results.**

| True Condition | Predicted Condition | |
|---|---|---|
| | Negative | Positive |
| Negative | True Negative(TN) | False Positive (FP) |
| Positive | False Negative (FN) | True Positive (TP) |

## SHAP analysis

To provide insight into our model's decision-making process, we employed SHAP analysis. SHAP, a unified framework for interpreting predictions, was introduced in 2017 as the only consistent and locally accurate feature attribution method based on expectations [49]. In cooperative game theory, SHAP values are computed as follows:

$$\phi_i = \sum_{S \subseteq F \setminus \{i\}} \frac{|S|!(|F|-|S|-1)!}{|F|!} [f_{S \cup \{i\}}(x_{S \cup \{i\}}) - f_S(x_S)]$$

(5)

## Results

This section presents the experimental results, which are organized into three main parts: (1) classifier performance rankings, (2) SHAP-based feature importance analysis, and (3) feature reduction outcomes based on global SHAP rankings. These findings were derived from the methodological pipeline illustrated in Fig 2, which systematically processes the input translations through dependency parsing, feature engineering, model training, SHAP interpretation, and feature reduction.

## Model performance

Using the full dataset with 776 samples, we conducted a 10-fold cross-validation to evaluate the performance of the 16 classifiers. Our evaluation shows that dependency-based syntactic features consistently help distinguish human and LLM translations.The comprehensive evaluation revealed consistent discriminative capability in distinguishing human and LLM translations using the dependency triplet features. As shown in Table 4, SVM achieved the highest mean F1-score (0.93 ± 0.02), demonstrating both superior performance and stability across folds. The top-performing models (SVM, MLP, and CatBoost) all exceeded a 0.90 mean F1-score, indicating that syntactic patterns captured by dependency triplets are robustly detectable across different algorithmic approaches.

## SHAP results

**Global feature contribution.** To comprehensively assess the influence of dependency triplet features across different modeling approaches, we performed SHAP analysis on the top and last 3 performing models. The results revealed both consistent patterns and model-specific variations in feature importance, as shown in Table 5 and Table 6:

Two clear patterns emerged from the SHAP analysis across all six models. First, *aux_VBD_VBN, nsubj_PRP_VBD* and *prt_RP_VBD* consistently ranked among the most important features in both the top and last three models, demonstrating their fundamental role in distinguishing translations. While the top three models shared six common syntactic features, the last three models showed weaker convergence—only K-Nearest retained five of these features, while Naive Bayes and Decision Tree each preserved just three.

Despite this shared core, model architectures significantly influenced feature weighting. The top models prioritized similar syntactic patterns (e.g., *aux_VBD_VBN*, *prt_RP_VBD*), whereas CatBoost uniquely emphasized *det_DT_NN*, reflecting its tree-based design. Among weaker models, Decision Tree amplified this *det_DT_NN* focus to an extreme (SHAP = 0.157), while K-Nearest and Naive Bayes all emphasized different syntactic features other than the shared two features. This divergence suggests that while core syntactic features remain universally important, architectural biases—especially in less robust models—distort their relative interpretative significance.

For further detailed SHAP visualization and feature reduction analysis in Section Feature reduction, we selected SVM due to its best performance. The beeswarm plot (Fig 3) shows that *aux_VBD_VBN* as the most influential feature, with higher feature values (red) strongly correlating with negative SHAP values, indicating their association with human

**Input:** Parallel corpus C (human/LLM translations)
**Output:** Classification metrics | SHAP analysis | Feature reduction curves

```
1. Dependency Parsing
    for T in C:
        doc ← spaCy_en(T)      # T stands for each translation sample
        extract (dep_rel, dep_POS, head_POS) from doc
        store as triplets
2. Feature Engineering
    triplet_freq ← count_frequency (unique_triplet)
    features ← filter(triplet_freq, min_freq=500)
    # normalized by text length with triplets in each T
    build dataset containing 108-dimensional feature vectors
3. Model Training
    classifiers ← {SVM, MLP, ..., DecisionTree}  # total: 16 models
    for clf in classifiers:
        perform 10-fold cross-validation on the dataset:
        compute mean F1 for clf
4. SHAP Interpretation
# SVM, MLP, Catboost & K-Nearest Neighbors, Naive Bayes, Decision Tree
    topLast3_models ← top & last 3 classifiers by mean F1
    for model in topLast3_models :
        explainer ← shap.KernelExplainer(model.predict_proba, background_data)
        shap_values ← explainer.shap_values(compu_data)
        global_imp ← sum(abs(shap_values), axis=0)
        top10_features ← argsort(global_imp)[-10:]
    # Visualize (mainly for SVM)
    plot_beeswarm(shap_values)
    plot_case_bars(shap_values[sample_cases])
    plot_dependence(top10_features)
5. Feature Reduction   # Visualize (mainly for SVM)
    ranked_features ← sort_by(global_imp, ascending=False)
    for k in [108, 98, ..., 8]:
        selected features ← ranked_features[:k]
        train model with 10-fold cross-validation using selected features
        record mean F1
    plot_f1_vs_feature_number()
```

**Fig 2. Dependency triplet processing workflow.**

translations (class 0). This pattern extends to other key features including *prt_RP_VBD*, *pobj_PRP_IN*, *nsubj_PRP_VBD*, and *poss_PRP$_NNS*. Conversely, features like *neg_RB_VB*, *det_DT_NN*, *acomp_JJ_VBZ* and *nsubj_NNP_VBD*, demonstrate the opposite relationship: higher values correspond to positive SHAP values, marking them as indicators of LLM

**Table 4. Classifier Performance Ranking.**

| Rank | Model | Mean F1 | Std F1 |
|------|-------|---------|--------|
| 1 | SVM | 0.93 | 0.02 |
| 2 | MLP | 0.92 | 0.03 |
| 3 | CatBoost | 0.91 | 0.04 |
| 4 | XGBoost | 0.90 | 0.03 |
| 5 | LightGBM | 0.89 | 0.03 |
| 6 | Gradient Boosting | 0.89 | 0.04 |
| 7 | Logistic Regression | 0.89 | 0.03 |
| 8 | LDA | 0.88 | 0.03 |
| 9 | Extra Trees | 0.88 | 0.03 |
| 10 | Random Forest | 0.88 | 0.02 |
| 11 | SGD | 0.87 | 0.02 |
| 12 | AdaBoost | 0.84 | 0.03 |
| 13 | QDA | 0.82 | 0.05 |
| 14 | K-Nearest Neighbors | 0.81 | 0.05 |
| 15 | Naive Bayes | 0.79 | 0.02 |
| 16 | Decision Tree | 0.74 | 0.05 |

**Table 5. Top 10 Features by Absolute SHAP Value Across the Top 3 Models.**

| Rank | SVM | SHAP | MLP | SHAP | CatBoost | SHAP |
|------|-----|------|-----|------|----------|------|
| 1 | **aux_VBD_VBN** | 0.047 | **pobj_PRP_IN** | 0.051 | det_DT_NN | 0.058 |
| 2 | **prt_RP_VBD** | 0.044 | **prt_RP_VBD** | 0.050 | **neg_RB_VB** | 0.055 |
| 3 | **pobj_PRP_IN** | 0.041 | **aux_VBD_VBN** | 0.048 | **aux_VBD_VBN** | 0.054 |
| 4 | **nsubj_PRP_VBD** | 0.040 | **neg_RB_VB** | 0.044 | **nsubj_PRP_VBD** | 0.046 |
| 5 | **neg_RB_VB** | 0.037 | **nsubj_PRP_VBD** | 0.041 | pobj_PRP_IN | 0.046 |
| 6 | det_DT_NN | 0.033 | mark_IN_VBD | 0.041 | **prt_RP_VBD** | 0.035 |
| 7 | **acomp_JJ_VBZ** | 0.028 | nsubj_NNP_VBD | 0.040 | **acomp_JJ_VBZ** | 0.031 |
| 8 | poss_PRP$_NNS | 0.028 | dobj_PRP_VBD | 0.040 | advcl_VBG_VBD | 0.027 |
| 9 | nsubj_NNP_VBD | 0.024 | **acomp_JJ_VBZ** | 0.037 | aux_VBD_VB | 0.025 |
| 10 | dobj_PRP_VBD | 0.023 | amod_JJ_NNS | 0.036 | pobj_NN_IN | 0.025 |

Note: The features that appear in the top 10 SHAP values across all **top three models** are marked bolded. Additionally, features that appear in the top 10 SHAP values for both the top three and the bottom three models are shown in <u>bold with underline</u>.

**Table 6. Top 10 features by absolute SHAP value across the last 3 models.**

| Rank | K-Nearest | SHAP | Naive Bayes | SHAP | Decision Tree | SHAP |
|------|-----------|------|-------------|------|---------------|------|
| 1 | **aux_VBD_VBN** | 0.020 | **aux_VBD_VBN** | 0.053 | det_DT_NN | 0.157 |
| 2 | **nsubj_PRP_VBD** | 0.017 | nsubj_PRP_VBN | 0.052 | **aux_VBD_VBN** | 0.082 |
| 3 | det_DT_NN | 0.015 | dobj_NN_VBZ | 0.039 | auxpass_VBD_VBN | 0.076 |
| 4 | **pobj_PRP_IN** | 0.015 | prep_IN_VBZ | 0.034 | dobj_PRP_VBD | 0.071 |
| 5 | dobj_PRP_VBD | 0.015 | **nsubj_PRP_VBD** | 0.030 | **prt_RP_VBD** | 0.043 |
| 6 | nsubj_PRP_VBN | 0.014 | **prt_RP_VBD** | 0.030 | aux_VBD_VB | 0.036 |
| 7 | mark_IN_VBD | 0.013 | nsubj_NN_VBZ | 0.027 | conj_VBD_VBD | 0.034 |
| 8 | pcomp_VBG_IN | 0.012 | pobj_PRP_IN | 0.027 | advmod_RB_VBD | 0.033 |
| 9 | **prt_RP_VBD** | 0.012 | cc_CC_VBZ | 0.026 | **nsubj_PRP_VBD** | 0.029 |
| 10 | **acomp_JJ_VBZ** | 0.012 | prep_IN_NN | 0.025 | relcl_VBD_NN | 0.028 |

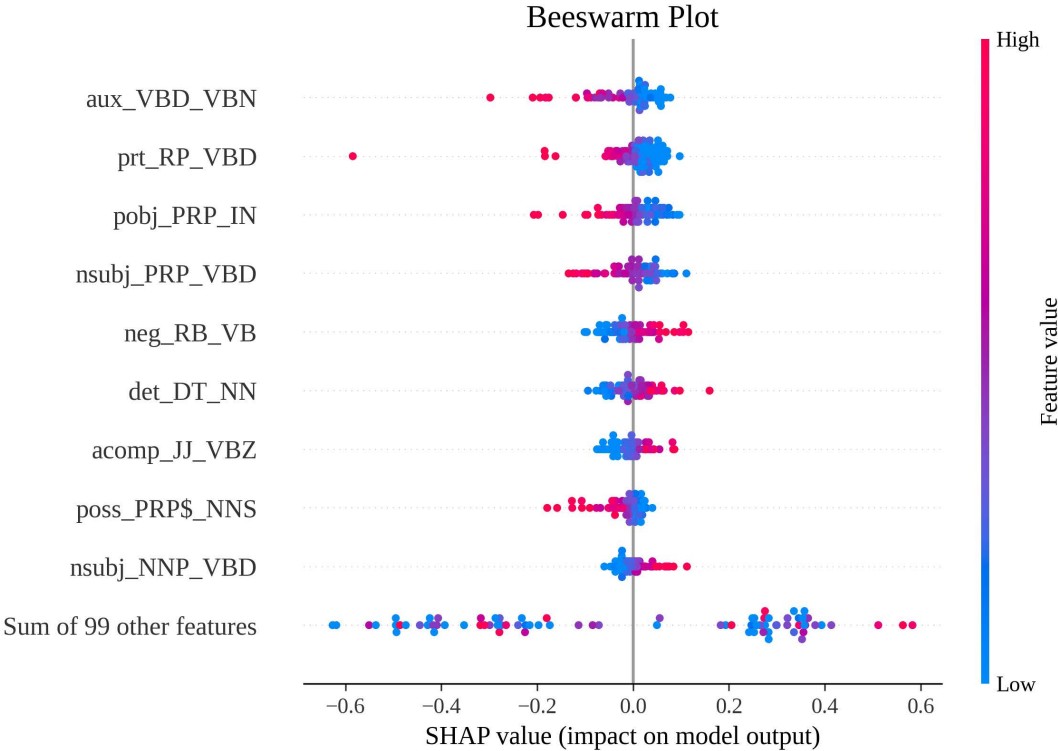

**Fig 3. Beeswarm plot of global feature contribution.**

translations (class 1). Importantly, this reciprocal relationship holds consistently – features predictive of one class at high values become indicators of the opposite class at low values (e.g., lower *aux_VBD_VBN* indicates class 1; lower *neg_RB_VB* indicates class 0). This clear bifurcation in feature impacts highlights how distinct syntactic patterns systematically drive classification decisions in opposite directions.

## Local feature contribution

Fig 4 shows how key features influence classification decisions locally: Sample 1 is predicted as class 1 (LLM) due to the mostly positive contributions (except *mark_IN_VBD*) from its top 9 features and a positive sum from all other features, while Sample 2 also favors class 1 with mostly positive influences (except *poss_NN_NN*). In contrast, Sample 3 is classified as class 0 (human) because only *nsubj_NNP_VBD* and *advmod_RB_VBG* are positive, whereas all other top features and the overall sum are strongly negative.

Besides, the analysis confirms consistent feature behavior across global and local contexts. Features like *neg_RB_VB*, *nsubj_NNP_VBD*, *acomp_JJ_VBZ*, and *det_DT_NN* maintain positive correlations (higher/lower feature values→higher/lower SHAP→class 1/class 0) in both global importance and individual samples, while *aux_VBD_VBN*, *prt_RP_VBD*, *pobj_PRP_IN*, and *nsubj_PRP_VBD* consistently show negative correlations (higher/lower feature values→lower/higher SHAP→class 0/class1). This alignment validates these dependency patterns as reliable indicators for translation classification.

## Dependence plot analysis

To examine whether the most important features identified by the SHAP analysis are influenced by interactions with other features, we generated dependence plots for the top dependency triplet features (Fig 5). These plots display the relationship

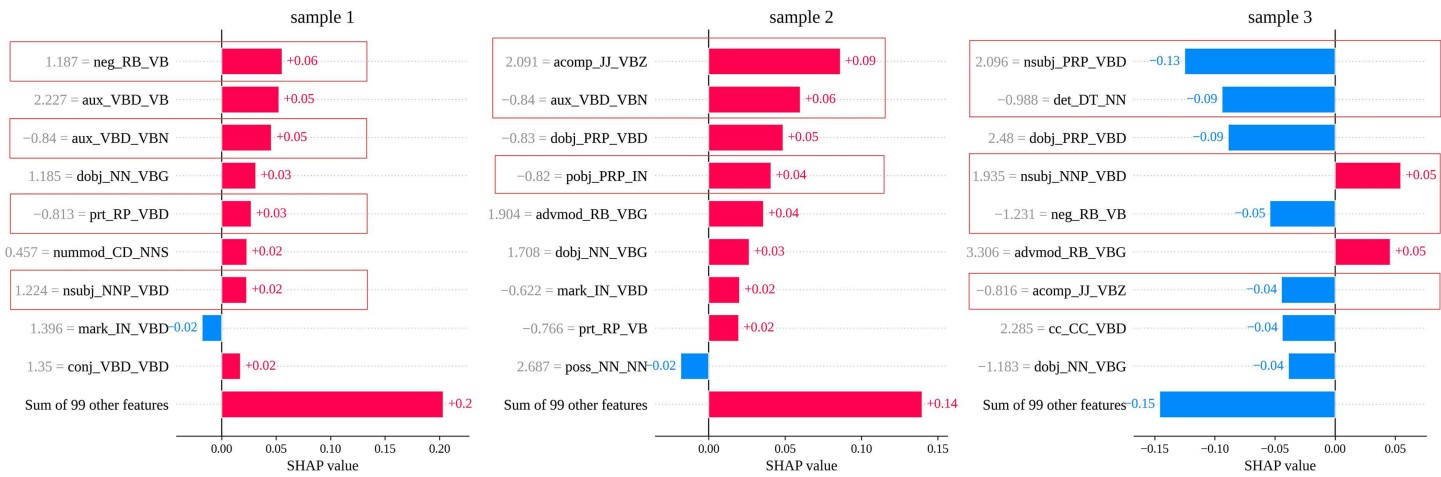

**Fig 4. SHAP value bar plot for three specific samples.**

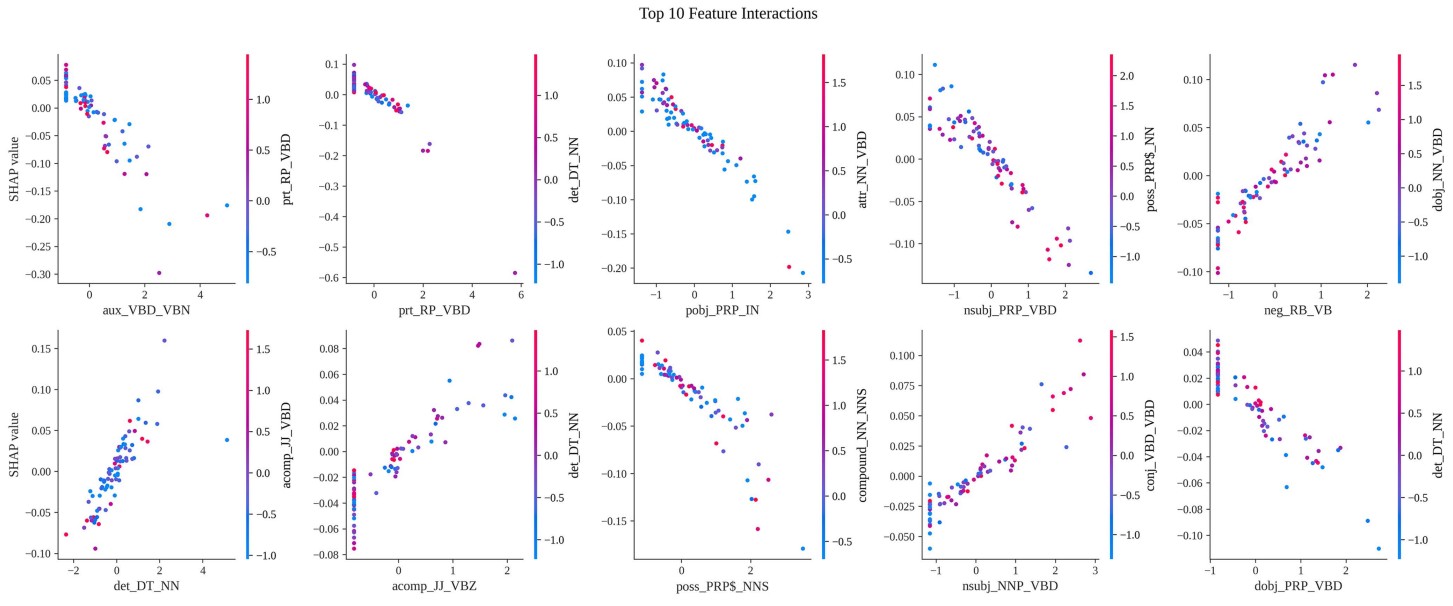

**Fig 5. Dependence plots for top features.**

between each feature's value and its corresponding SHAP value, with colors representing the values of interacting features. Upon examining these plots, we observed that the SHAP values for the highest-ranking features, such as *aux_VBD_VBN* and *prt_RP_VBD*, primarily correlate with their own feature values without significant influence from interacting features. The uniform color distribution across the feature values indicates that interactions have a minimal impact on the SHAP values. This limited impact of feature interactions reinforces the validity of our feature importance rankings, confirming that the top dependency triplet features independently contribute most substantially to the model's ability to distinguish between human and LLM-generated translations. The effectiveness of SHAP-based feature selection is further validated, as it highlights the critical syntactic patterns that underpin machine translationese without being confounded by feature interactions.

## Feature reduction

Using 10-fold cross-validation with the best-performing SVM model, we evaluated the impact of feature reduction based on the global feature importance rankings derived from SHAP analysis. Starting with the full set of 108 dependency triplet features, we systematically removed the least important features in increments of 10. As shown in Fig 6, the results indicated that the model's performance remained stable when reducing the feature set to approximately 38 features (Table 7), with the F1 score consistently above 90%. This demonstrates that a significant portion of the features were redundant and that the most critical features as identified through SHAP, were sufficient for maintaining robust classification performance.

Notably, even when reducing the feature set to the top 8 most important features, the model still achieved an F1 score exceeding 80%. This encouraging result further underscores the effectiveness of the SHAP-based feature selection, confirming that the most influential features were correctly identified and essential for distinguishing between human and LLM-generated translations. This streamlined approach not only simplifies the model but also ensures that it remains robust and reliable in practical settings.

## Discussion

The findings of this study demonstrate meaningful progress in the field of text classification, particularly in distinguishing between human translations and LLM-generated translations through the use of dependency triplet features. By integrating dependency syntax and part-of-speech combinations, this research not only achieved robust classification

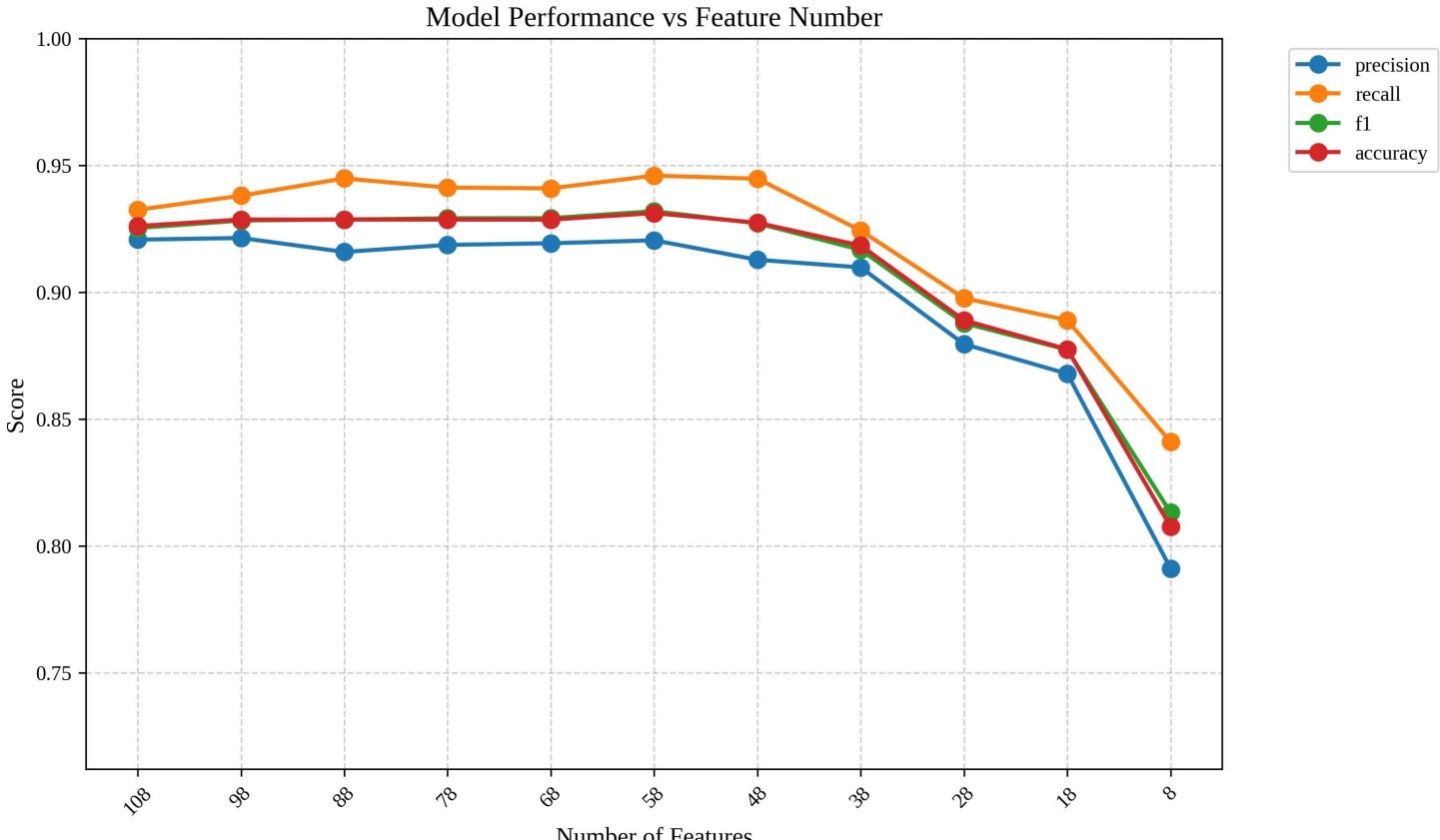

**Fig 6. Model performance during feature reduction.**

**Table 7. Top 38 dependency triplets by absolute SHAP value.**

| Feature | Value | Feature | Value | Feature | Value |
|---|---|---|---|---|---|
| aux_VBD_VBN | 0.047 | prt_RP_VBD | 0.044 | pobj_PRP_IN | 0.041 |
| nsubj_PRP_VBD | 0.040 | neg_RB_VB | 0.037 | det_DT_NN | 0.033 |
| acomp_JJ_VBZ | 0.028 | poss_PRP$_NNS | 0.028 | nsubj_NNP_VBD | 0.024 |
| dobj_PRP_VBD | 0.023 | nummod_CD_NNS | 0.023 | mark_IN_VBD | 0.023 |
| ccomp_VBD_VBD | 0.023 | dobj_NN_VBG | 0.020 | amod_JJ_NNS | 0.020 |
| nsubj_PRP_VBN | 0.020 | aux_VBD_VBG | 0.019 | advmod_RB_RB | 0.019 |
| advcl_VBG_VBD | 0.019 | prt_RP_VB | 0.019 | advmod_RB_VB | 0.019 |
| dobj_PRP_VB | 0.018 | cc_CC_NN | 0.018 | agent_IN_VBN | 0.017 |
| cc_CC_JJ | 0.017 | aux_VBD_VB | 0.016 | acomp_JJ_VBP | 0.016 |
| dobj_NN_VB | 0.015 | relcl_VBD_NN | 0.015 | conj_VBD_VBD | 0.014 |
| compound_NN_NN | 0.014 | nsubj_NNP_VB | 0.014 | compound_NNP_NNP | 0.013 |
| cc_CC_VB | 0.013 | prep_IN_VBZ | 0.013 | poss_PRP$_NN | 0.013 |
| det_DT_NNS | 0.012 | pobj_NN_IN | 0.012 | | |

performance but also provided deeper insights into the distinctive linguistic patterns of LLM translations. This study marks an important expansion of translationese research by shifting the traditional focus on distinguishing original texts from human translations to the emerging domain of machine translationese, specifically the differences between human translations and LLM-generated outputs, thereby broadening the scope of inquiry in this field. Methodologically, the study further enhanced its methodological rigor by SHAP analysis for interpretability, and feature reduction techniques for model streamlining. These improvements are supported by the stable classification results, underscoring the efficacy of the proposed approach and its broader implications for translation studies and natural language processing.

The study achieved a peak classification performance of 93% F1-score (SVM), with all 16 models demonstrating stable discrimination, as highlighted in Section Model performance. This performance surpasses other models that relied on entropy or n-gram-based features for translationese detection. For instance, Liu et al. [37] reported achieving a best accuracy of 84.3% using entropy values derived from character, wordform, and POS n-grams. Similarly, Baroni and Bernardini [12] achieved 86.7% accuracy with word and POS n-grams, while Wang et al. [14] attained 88.5% accuracy using syntactic tree and entropy-based methods. Notably, these studies utilized larger datasets, such as the ZCTC and LCMC corpora, where each sample contained approximately 2000 words. In contrast, our study achieved better classification performance with significantly smaller samples, averaging fewer than 400 words per sample. The results underscore the efficiency and robustness of our methodology, highlighting its potential as an effective tool for translationese detection and analysis in natural language processing tasks.

A key contribution of this study lies in its innovative use of dependency triplet features, which integrate dependency relations with part-of-speech tags to construct a rich and linguistically interpretable feature set. This approach contrasts with traditional approaches that typically rely on isolated or surface-level features such as type-token ratio (TTR), lexical density, and word frequencies. These conventional features often capture only superficial aspects of texts, whereas our method captures deeper syntactic patterns that are more reflective of translation-specific traits. Prior studies, including those by Liang and Sang [13] and Wang et al. [14], have noted that traditional metrics often obscure the complexity of translated language and produce inconsistent results across language pairs. Our findings demonstrate the effectiveness of this approach, as evidenced by the high classification accuracy and interpretability of the model.

Another important contribution of this study is the interpretability framework developed through SHAP analysis—a technique still rarely applied in translation research, with only Liu et al. [37] and Wang et al. [14] employing it for global feature analysis. This framework quantified the contribution of individual dependency triplet features at both global and local

levels, revealing their interaction effects in distinguishing human and LLM-generated translations. Key patterns emerged across the top and last 3 performing models: *aux_VBD_VBN, nsubj_PRP_VBD* and *prt_RP_VBD* consistently ranked among the most important features, demonstrating their fundamental role in classification. While the top-performing models shared six core syntactic features (e.g., *aux_VBD_VBN*, *prt_RP_VBD*), weaker models exhibited architectural biases and less convergence—such as Decision Tree's exaggerated focus on *det_DT_NN*—highlighting how model design influences feature interpretability.

The analysis also identified distinct syntactic traits: higher values of *aux_VBD_VBN* strongly correlated with human translations (class 0), whereas features like *neg_RB_VB* and *det_DT_NN* were indicative of LLM outputs (class 1). This divergence underscores the unique syntactic signatures of LLM machine translationese. Crucially, the beeswarm plot confirmed a reciprocal relationship—low values of human-predictive features (e.g., *aux_VBD_VBN*) signaled LLM translations, and vice versa. Dependence plots further validated that these features operated primarily through direct values, with minimal interaction effects, reinforcing the reliability of SHAP rankings. Importantly, these patterns held consistently across global and local contexts: features like *neg_RB_VB* maintained positive SHAP correlations with LLM translations in both aggregate and individual predictions, while *aux_VBD_VBN* and *prt_RP_VBD* consistently aligned with human translations. This dual-level consistency solidifies the framework's robustness in decoding syntactic discriminators, even as model-specific biases modulate their relative emphasis.

Another critical key outcome of this interpretability framework was its application to feature reduction. Using SHAP-derived global feature importance, we were able to systematically reduce the feature set from 108 dependency triplets to just 38, while maintaining an F1 score of about 90%. Remarkably, even with only the top 8 features, the model achieved an F1 score exceeding 80%, demonstrating that the most influential dependency triplets identified by SHAP analysis were sufficient for robust classification. This reduction not only streamlined the model but also highlighted the redundancy of many features, further validating the effectiveness of the selected dependency triplets.

While this study demonstrates the effectiveness of dependency triplet features, it is important to acknowledge its limitations to contextualize the findings and guide future research. Our analysis was conducted on a single literary work (*Bian Cheng*) within one language pair (Chinese-English). This focused approach allowed for controlled comparisons but may limit the generalizability of our findings across different genres, text types, and language pairs. The syntactic patterns identified as characteristic of LLM translationese may vary when analyzing technical or journalistic texts, or when examining translations between languages with different typological characteristics. Therefore, future work should prioritize validating these features on a more diverse corpus to confirm their broader applicability.

From a methodological perspective, several avenues for future exploration also exist. Our work intentionally focused on interpretable linguistic features, and the strong, consistent performance across 16 different classifiers validates their discriminative power. A valuable next step would be to benchmark these feature-based models against end-to-end deep learning systems, such as fine-tuned versions of BERT or Llama. Furthermore, to build upon our robust cross-validation results, future studies could employ more complex evaluation schemes, such as a three-way split or nested cross-validation, especially to test the generalizability of interpretations derived from methods like SHAP on completely unseen data. Such research would further solidify the reliability of these syntactic features in real-world applications.

## Conclusion

In conclusion, this study advances the understanding of machine translationese by integrating dependency grammar with part-of-speech tagging and SHAP-based interpretability frameworks. This novel approach not only achieves robust classification performance but also provides deeper insights into the linguistic patterns distinguishing human and LLM-generated translations. While the findings are based on a specific genre and language pair, they highlight the potential of syntactic feature engineering in translation studies. Future research should aim to validate these findings across more diverse corpora and through broader methodological explorations, including benchmarking against deep learning models

under rigorous evaluation schemes. Such efforts could offer valuable implications for both theoretical advancements and practical applications in natural language processing. By doing so, they could foster the development of more sophisticated and accurate translation models, ultimately benefiting global communication efforts and enhancing our understanding of machine-generated text.

## Acknowledgments

We would like to thank the reviewers and the editor for their comments and suggestions. We would like to thank the anonymous reviewers and the editor for their comments and suggestions.

## Author contributions

**Conceptualization:** Shukang Zhang.

**Data curation:** Shukang Zhang.

**Formal analysis:** Shukang Zhang.

**Funding acquisition:** Chaoyong Zhao.

**Investigation:** Shukang Zhang.

**Methodology:** Shukang Zhang.

**Project administration:** Chaoyong Zhao.

**Resources:** Chaoyong Zhao.

**Software:** Shukang Zhang.

**Supervision:** Chaoyong Zhao.

**Visualization:** Shukang Zhang.

**Writing – original draft:** Shukang Zhang.

**Writing – review & editing:** Chaoyong Zhao.

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
