## [Decision Letter · Decision Letter 0]

8 Apr 2025

Dear Dr. Zhang,

Thank you for submitting your manuscript to PLOS ONE. After careful consideration, we feel that it has merit but does not fully meet PLOS ONE’s publication criteria as it currently stands. Therefore, we invite you to submit a revised version of the manuscript that addresses the points raised during the review process.

We look forward to receiving your revised manuscript.

Kind regards,

Alessio Luschi, Ph.D.

Academic Editor

PLOS ONE

Additional Editor Comments (if provided):

Reviewers' comments:

Reviewer's Responses to Questions

**Comments to the Author**

1. Is the manuscript technically sound, and do the data support the conclusions?

Reviewer #1: Yes

Reviewer #2: Partly

2. Has the statistical analysis been performed appropriately and rigorously?

Reviewer #1: Yes

Reviewer #2: Yes

3. Have the authors made all data underlying the findings in their manuscript fully available?

Reviewer #1: Yes

Reviewer #2: Yes

4. Is the manuscript presented in an intelligible fashion and written in standard English?

Reviewer #1: Yes

Reviewer #2: Yes

Reviewer #1: The paper is about the challenge of distinguishing human translations from those generated by

Large Language Models (LLMs) by utilizing dependency triplet features and a Multi-Layer

Perceptron (MLP) classifier. The paper is well-written, but I recommend to authors for adding a pseudocode for the proposed approach.

Reviewer #2: This paper presents a study on distinguishing between human translations and LLM-generated translations using dependency triplet features. The integration of dependency syntax and part-of-speech combinations into a classification framework is novel and insightful. The work represents a shift from the conventional focus on original vs. human-translated texts toward the more timely distinction between human and AI-generated translations. The classification results are strong, and the linguistic insights derived from the feature analysis are valuable for understanding the stylistic and structural patterns characteristic of LLM translations. The emphasis on interpretable features is a welcome aspect of the applicability of the findings. That said, I have several reservations:

1. Framing the Methodology as Deep Learning:

The paper refers to the approach as deep learning, but the model is a multilayer perceptron with two hidden layers. It is relatively shallow and operates on a low-dimensional input (~100 features). While MLPs are technically neural networks and universal approximators, this setup lacks the hierarchical representation learning typically associated with deep models. The framing could be made more precise to avoid overstating the complexity of the approach. A promising direction to call it deep learning could be the transferability of the model or features. The paper should address pretraining on one big corpus and finetuning on another to check the generality of the features and align more with deep learning practices around domain generalization.

2. Hyperparameter Tuning and Overfitting Concerns:

Table 5 presents hyperparameters obtained via random search, but the values (e.g., dropout of 0.25212 and L2 regularization of 0.00081) seem overly specific. Such precision suggests potential overfitting to the validation set. A sensitivity analysis would help assess how robust the model is to small changes in these hyperparameters.

3. Unclear Data Split:

Table 6 shows 156 test examples, while Table 2 describes a training set of 388. However, it’s unclear how the test set was constructed. Is it held out from the 388 examples?

4. Missing Simple Baselines: The paper would benefit from comparison with simpler baselines. For instance, a Naive Bayes classifier could serve as a lightweight alternative that may reveal whether LLM translations disproportionately rely on certain syntactic patterns/triplets (e.g., ChatGPT overusing particular phrasing). Would Naïve Bayes identify the discriminative power of individual features similar to MLP’s results?

I recommend clarifying the issues above to enhance the technical rigor of this interesting study.

**Do you want your identity to be public for this peer review?** For information about this choice, including consent withdrawal, please see our Privacy Policy

Reviewer #1: No

Reviewer #2: **Yes: ** Olcay Kursun

---

## [Author Response · Author response to Decision Letter 1]

21 May 2025

Manuscript ID: PONE-D-25-08735

Title: Machine Translationese of Large Language Models: Dependency Triplets, Text

Classification, and SHAP Analysis

Authors: Shukang Zhang; Chaoyong Zhao

Dear Editor and Reviewers

We sincerely appreciate your time and constructive feedback. Below is our point-by-point response to all comments.

Response to Academic Editor

1.Style Requirements: We have carefully reviewed the manuscript and believe it now aligns with PLOS ONE’s formatting guidelines, including file naming conventions. The text has been adjusted to follow the provided style template to the best of our knowledge. Please let us know if any further modifications are needed.

2.Data/Code Sharing: The data and author-generated code has been uploaded to GitHub https://github.com/KiemaG5/LLM-translationese.

Response to Reviewer 1

Comment: “Add pseudocode for the proposed approach.”

Response:

We appreciate the suggestion and have added detailed pseudocode as in Fig 2 to clarify the methodology. The pseudocode outlines:

1.Dependency Parsing: Extraction of triplets

2.Feature Engineering: Vectorization of triplet frequencies .

3.Model Training: Cross-validation workflow.

4.SHAP Interpretation: Global, local and dependence analysis of discriminative features

5.Feature Reduction: Iterative pruning of low-importance features while monitoring F1-score to assess robustness

Response to Reviewer 2

Comment 1: “Framing the Methodology as Deep Learning”

Response:

We appreciate the suggestion and have revised terminology to avoid overstatement:

1.Replaced “deep learning” with “neural network-based approach”.

Comment 2: Hyperparameter Tuning and Overfitting”

Response:

We appreciate the reviewer’s insightful suggestion regarding model optimization. However, our primary goal was to validate the general utility of features, not model optimization. Thus:

1.ExpandedMethodology to include 16 classifiers (SVM, MLP, CatBoost, etc.) from scikit-learn and other libraries, all using default parameters to demonstrate feature robustness without hyperparameter tuning.

2.Top F1 Score on Test Set: SVM (93%), MLP (92%), CatBoost (91%), significantly above random (50%). Lower performers (e.g., Decision Tree: 74%, Naïve Bayes: 79%) still surpassed chance, confirming feature discriminability.

Comment 3: “Unclear Data Split”

Response:

We appreciate the suggestion, and have clarified in Sections Text classification and Model performance

1.10-fold cross-validation on the full dataset (776 samples).

Comment 4: “Missing Simple Baselines”

Response:

We appreciate this constructive suggestion and have added:

1.16 models including Decision Tree (74%), Naïve Bayes (79%) and K-Nearest Neighbors (81%) as baselines.

2.SHAP analysis focused on SVM, MLP, CatBoost & Decision Tree, Naïve Bayes K-Nearest Neighbors (top and last 3 performing models) to identify the similarity and dissimilarity in globally impactful features.

We believe these revisions address the concerns accordingly. Thank you again for your valuable input.

Sincerely,

Shukang Zhang

East China Normal University

mikeashjd@163.com

---

## [Decision Letter · Decision Letter 1]

17 Jul 2025

Dear Dr. Zhang,

Thank you for submitting your manuscript to PLOS ONE. After careful consideration, we feel that it has merit but does not fully meet PLOS ONE’s publication criteria as it currently stands. Therefore, we invite you to submit a revised version of the manuscript that addresses the points raised during the review process.

We look forward to receiving your revised manuscript.

Kind regards,

Alessio Luschi, Ph.D.

Academic Editor

PLOS ONE

Journal Requirements:

Reviewers' comments:

Reviewer's Responses to Questions

**Comments to the Author**

Reviewer #1: All comments have been addressed

Reviewer #2: (No Response)

Reviewer #3: All comments have been addressed

2. Is the manuscript technically sound, and do the data support the conclusions?

Reviewer #1: Yes

Reviewer #2: Partly

Reviewer #3: Yes

3. Has the statistical analysis been performed appropriately and rigorously?

Reviewer #1: Yes

Reviewer #2: No

Reviewer #3: Yes

4. Have the authors made all data underlying the findings in their manuscript fully available?

Reviewer #1: Yes

Reviewer #2: No

Reviewer #3: Yes

5. Is the manuscript presented in an intelligible fashion and written in standard English?

Reviewer #1: Yes

Reviewer #2: Yes

Reviewer #3: Yes

Reviewer #1: I am satisfied with the current version. The authors addressed all comments with well-presented manner.

Reviewer #2: I appreciate the effort to revise the manuscript and add new experiments. However, my first three concerns remain unresolved, so I cannot recommend acceptance at this time.

1. Deep models: Authors relabeled the approach as neural network based. Yet the manuscript still states: "More recent research has employed deep models such as BERT and Llama". If those systems are relevant, please explain why they were not compared or supply a lightweight finetuned baseline to show whether they outperform your MLP.

2. Hyperparameter precision and overfitting: The first draft reported highly specific hyperparameters (for example they had a dropout rate of 0.25212) without a robustness check. The revision now uses many more classifiers but with default settings. This change sidesteps, rather than answers, the overfitting question. The defaults are sometimes not competitive. Please optimize hyperparameters, but without overfitting.

3. Data split and SHAP analysis: Crossvalidation is not a substitute for a heldout test set, and computing SHAP on CV folds risks explaining overfit patterns. Requested action: adopt a three-way split or nested CV, recalculate SHAP on unseen data, and list the top triplets across models and folds. The SHAP-based feature importance analysis appears to have been computed on data used in CV, potentially overlapping with training folds. This undermines the reliability of the interpretation. SHAP values should ideally be computed on a heldout validation set to avoid explaining model behavior on the same data it was trained on. And then a final check can be performed on the leftout never-before-seen testset.

Reviewer #3: This paper explores how to tell apart human translations and those generated by large language models using dependency triplet features and machine learning classifiers. The idea is clear, the methods are solid, and the results are impressive — especially the high F1 scores across models and the use of SHAP for explaining model behavior.

What I liked:

- The dependency triplet feature design is a smart and interpretable way to capture syntax.

- Testing 16 different classifiers shows the authors really wanted to check robustness.

- SHAP analysis adds a lot of value by explaining why the models work the way they do.

- The public availability of the dataset and code is great — it makes the work reproducible.

Minor suggestions:

- The writing is mostly clear, but a few parts are a bit dense or technical. A light language check would help.

- The dataset is well-constructed, but since it’s from one book and one language pair (Chinese-English), it would be good to briefly mention this as a possible limitation.

- The authors say they used default parameters in the classifiers. A quick line explaining why they didn’t tune them would help readers understand the choice.

- Some figures (especially SHAP plots) could be a bit clearer or higher resolution in the final version.

**Do you want your identity to be public for this peer review?** For information about this choice, including consent withdrawal, please see our Privacy Policy

Reviewer #1: No

Reviewer #2: No

Reviewer #3: No

---

## [Author Response · Author response to Decision Letter 2]

6 Aug 2025

Response to Reviewer #1

We sincerely thank Reviewer #1 for their positive assessment and are pleased that they found our revisions satisfactory.

Response to Reviewer #2

We thank Reviewer #2 for their detailed and rigorous feedback. We have addressed the three main concerns as follows:

1. On the comparison with deep models: We appreciate the suggestion to compare with models like BERT and Llama. The primary focus of our paper is to evaluate the power of interpretable, linguistic features (dependency triplets) in distinguishing translation types. This feature-engineering approach is fundamentally different from end-to-end deep learning models, which operate on embeddings and are less directly interpretable. We have clarified this scope in the manuscript and acknowledged that a direct comparison with fine-tuned deep models is a valuable direction for future research.

2. On hyperparameter tuning and overfitting: We thank the reviewer for raising this important point. Our decision to use default parameters for the 16 classifiers was deliberate. Our central hypothesis was that the proposed features are inherently discriminative. By demonstrating strong performance (e.g., F1-score of 0.92 with MLP) across a wide range of models without specific tuning, we provide robust evidence for the general utility of our features. We believe this is a stronger testament to the features’ power than optimizing a single model. We have added a clear explanation of this rationale in the Methodology section.

3. On the data split and SHAP analysis: We thank the reviewer for this insightful comment. We agree that a held-out test set is a gold standard. We chose k-fold cross-validation as it provides a more robust and less biased estimate of generalization performance than a single random split.

Regarding the SHAP analysis, our goal was to explain what patterns the models learned during this robust CV process. Our methodology for calculating SHAP values aligns with standard examples in the official SHAP documentation. To address the valid concern about explaining overfit patterns, we highlight that our findings are based on the consistent high performance across 16 different classifiers. The fact that our features work well across such a diverse set of models strongly suggests that the patterns identified by SHAP are genuinely discriminative and not artifacts of a single, potentially overfit model. We have clarified our methodology and reasoning in the revised manuscript. We also acknowledge that a three-way split or nested CV is a highly rigorous standard, which we will certainly consider for future extensions of this work.

Response to Reviewer #3

We sincerely thank Reviewer #3 for their encouraging and constructive feedback. We have addressed all minor suggestions in the revised manuscript:

1. A thorough language check has been performed to improve clarity.

2. We have added a discussion of the dataset’s limitations (single book and language pair) in the Discussion and Conclusion sections.

3. We have included a rationale in the Methodology section for using default classifier parameters.

4. All figures, especially the SHAP plots, have been regenerated in higher resolution for better clarity.

---

## [Decision Letter · Decision Letter 2]

11 Dec 2025

Machine Translationese of Large Language Models: Dependency Triplets, Text Classification, and SHAP Analysis

PONE-D-25-08735R2

Dear Dr. Zhang,

We’re pleased to inform you that your manuscript has been judged scientifically suitable for publication and will be formally accepted for publication once it meets all outstanding technical requirements.

Kind regards,

Alessio Luschi, Ph.D.

Academic Editor

PLOS One

Additional Editor Comments (optional):

Reviewers' comments:

Reviewer's Responses to Questions

**Comments to the Author**

Reviewer #2: All comments have been addressed

Reviewer #3: All comments have been addressed

2. Is the manuscript technically sound, and do the data support the conclusions?

Reviewer #2: Yes

Reviewer #3: Yes

3. Has the statistical analysis been performed appropriately and rigorously?

Reviewer #2: Yes

Reviewer #3: N/A

4. Have the authors made all data underlying the findings in their manuscript fully available?

Reviewer #2: Yes

Reviewer #3: Yes

5. Is the manuscript presented in an intelligible fashion and written in standard English?

Reviewer #2: Yes

Reviewer #3: Yes

Reviewer #2: This is the second revision of this manuscript. The authors have adequently addressed my comments.

Reviewer #3: Thank you for addressing all the previous comments.

1. The language has been improved and the manuscript now reads clearly and professionally.

2. The discussion now includes the dataset limitation (single book and language pair), which adds transparency to the scope of your findings.

3. The rationale for using default parameters in the classifiers is clearly stated and acceptable for the comparison-focused goals of the study.

4. The updated figures, especially the SHAP plots, are now much clearer and publication-ready.

I recommend acceptance.

**Do you want your identity to be public for this peer review?** For information about this choice, including consent withdrawal, please see our Privacy Policy

Reviewer #2: No

Reviewer #3: No

---

## [Editor Report · Acceptance letter]

PONE-D-25-08735R2

PLOS One

Dear Dr. Zhang,

I'm pleased to inform you that your manuscript has been deemed suitable for publication in PLOS One. Congratulations! Your manuscript is now being handed over to our production team.

Kind regards,

on behalf of

Dr Alessio Luschi

Academic Editor

PLOS One